# Comparative Analysis of the Microbiome across the Gut–Skin Axis in Atopic Dermatitis

**DOI:** 10.3390/ijms22084228

**Published:** 2021-04-19

**Authors:** Dong Hoon Park, Joo Wan Kim, Hi-Joon Park, Dae-Hyun Hahm

**Affiliations:** 1College of Medicine, Kyung Hee University, Seoul 02447, Korea; dhpark05@khu.ac.kr (D.H.P.); joowan0509@naver.com (J.W.K.); 2Acupuncture and Meridian Science Research Center, Kyung Hee University, Seoul 02447, Korea; acufind@khu.ac.kr; 3Department of Biomedical Sciences, Graduate School, Kyung Hee University, Seoul 02447, Korea; 4BioNanocomposite Research Center, Kyung Hee University, Seoul 02447, Korea

**Keywords:** atopic dermatitis, skin microbiome, gut microbiome, gut–skin axis

## Abstract

Atopic dermatitis (AD) is a refractory and relapsing skin disease with a complex and multifactorial etiology. Various congenital malformations and environmental factors are thought to be involved in the onset of the disease. The etiology of the disease has been investigated, with respect to clinical skin symptoms and systemic immune response factors. A gut microbiome–mediated connection between emotional disorders such as depression and anxiety, and dermatologic conditions such as acne, based on the comorbidities of these two seemingly unrelated disorders, has long been hypothesized. Many aspects of this gut–brain–skin integration theory have recently been revalidated to identify treatment options for AD with the recent advances in metagenomic analysis involving powerful sequencing techniques and bioinformatics that overcome the need for isolation and cultivation of individual microbial strains from the skin or gut. Comparative analysis of microbial clusters across the gut–skin axis can provide new information regarding AD research. Herein, we provide a historical perspective on the modern investigation and clinical implications of gut–skin connections in AD in terms of the integration between the two microbial clusters.

## 1. Introduction

Atopic dermatitis (AD) is a skin disorder characterized by itching, scratching, inflammation and lichenification; the symptoms usually reoccur after remission and exacerbate over time. AD patients have a genetic predisposition to defective skin, which can increase the risk of environmental allergens [1]. In recent years, many AD patients have shown to have defects in epidermal cornification and skin barrier function due to abnormalities in the filament-aggregating protein gene such as filaggrin. In addition to genetic factors, environmental factors such as climate, air pollutants, diet, irritant exposure, and breastfeeding are implicated in AD outbreaks [2]. These environmental factors change the skin epidermal environment or disrupt the immune function, making patients vulnerable to AD. However, the mechanism underlying the development of AD is complicated, and the AD outbreaks cannot be attributed to any single genetic, immunological, or environmental factor. With the development of culture-free techniques, researchers have shown that microbes, both on the skin and in the gut, may influence the course of AD. Although antiseptic therapy has been used since decades for treating AD, the traditional culture-based methods and modern metagenomics have led to the development of targeted treatment of microbial dysbiosis in the gut or skin as an integrated treatment plan for AD in the future. Hence, researchers have attempted to determine what microbes in the skin and intestine are related to AD and how they can be utilized to treat AD [3].

The skin and associated immune system are primarily dominated by several skin microbes. The dysfunction of the adaptive immune response and epidermal barrier which is responsible for AD is closely related to the skin microbes. Previously, researchers attempted to identify the causes of AD based on not only genetics and heredity but also the imbalance of resident skin microbes or dominance of certain microbial clusters. 

The importance of the gut microbiome has been highlighted during birth in humans, as the immune system is exposed to various microbes and continues to develop since the fetal stage up to 7-8 years; many researchers have focused on the causality of various diseases related to intestinal immunity, rendering it inevitable to discuss the role of the gut microbiome in AD [4,5]. However, understanding how the intestinal microbiome controls the environment of the peripheral tissues such as the skin requires further research. It has been suggested that the imbalance of the gut microbiome affected the systemic immune system of the body and thus can exacerbate AD [3,6,7,8,9]. However, understanding the immunological, metabolic, and neurological interactions between the gut microbiome and skin is necessary to determine the exact mechanisms by which the gut microbiome causes AD and how it is involved in aggravating AD symptoms. Such interactions between the gut microbiome and other organs or tissues have long been investigated in the gut microbiome of the fruit fly model [10,11].

The “gut–brain–skin axis” is a theory that suggests that the gut microbial environment interacts with the brain and skin tissues, and plays an important role in the development of related diseases: autism spectrum disorders, Parkinson’s disease and Alzheimer’s disease in the brain, and psoriasis, psoriatic arthritis, Behcet’s disease and acne vulgaris in the skin [12,13]. The gut microbiome was found to act as a bridge between the immune system and the nervous system. In recent studies, in particular, this axis is used to describe the correlation between gut microbial communities, emotional states, and systemic and skin inflammation, and may be closely associated with the etiological mechanism between psoriasis and depression [14]. Using clinical cases of psoriasis and its animal models, important communication pathways have been identified along the axis associated with the regulation of neurotransmitters in the microbiome [15]. It can be therefore expected that a new strategy can be found to treat both psoriasis and depression based on the gut-brain-skin axis.

The roles of the gut and skin microbiomes in AD development are somewhat similar. The high diversity of the gut microbiome controls the immunity of the entire body, which increases the number of regulatory T cells (Tregs), short-chain fatty acids (SCFAs), and immune tolerance [4,16,17,18,19]. Conversely, the imbalance of the gut microbiome, reduced microbial diversity, domination of toxic microorganisms, and the absence of specific microorganisms increase the susceptibility of children to frequent secondary skin infections and immune-related diseases including AD [4,17,20]. Skin microbes also control local and systemic immune systems by generating antimicrobial peptides, a complementary system and the control of regulatory immune system, which can be induced to increase interleukin (IL)-1, IL-17A, IL-2 and interferon (IFN)-γ from the cluster of differentiation (CD)4^+^ forkhead box P3 (Foxp3)^+^ Tregs by stimulating the skin to increase the activity of Treg cells through toll-like receptors (TLRs) [3,21,22]. This review focuses on the gut–skin axis and attempts to understand how the gut and skin microbiomes interact with each other in AD. Determining the mutual effects of the gut microbiome and skin immune response in the causes of AD is important. Further to the attempt to simply transfer AD from the skin to the intestine, the interaction will be considered to broaden our knowledge of AD and human microbiome. These findings will be clinically important as they can change the way in which AD patients have been treated conventionally by using local steroids, local immunosuppressants (calcineurin antagonists) and anti-histamines. Furthermore, they can introduce new therapeutic methods associated with the microbiomes of the gut and skin. In this review, we summarize and interpret recent findings on the interaction of the gut and skin microbiomes associated with AD pathology, highlighting the novel role of the gut–skin axis for the management of AD.

## 2. The Role of the Skin Microbiome in Atopic Dermatitis

The skin is the first barrier to the external environment and effectively shields against bacterial infections [23]. The skin microbiota consists of up to 10^7^ microorganisms per square centimeter and mainly includes *Propionibacterium*, *Streptococcus*, *Staphylococcus* and *Corynebacterium* [24,25]. The composition of the skin microbiota varies remarkably across individuals in different environments. For example, *Propionibacterium* and *Staphylococcus* are mainly found in oil-rich areas, *Corynebacterium* and *Staphylococcus* in moist areas and β*-Proteobacterium* in dry areas [26,27,28]. Furthermore, the skin microbial ecosystem can vary depending on the nutritional status, health condition, age and environment of an individual. 

The skin microbial ecosystem interacts with various tissues and the immune system of the skin, and affects the health and functions of the body. Previous studies have shown that contact with maternal or external microorganisms plays an important role in the formation and maturation of immunity in infants [29,30]. In addition, the skin microbiome plays an important role in maintaining the skin barrier. The skin microbiome not only helps in the formation of mature keratinocytes in the stratum corneum, the outer layer of the skin barrier, but also controls the immune system of the entire body [31]. Microbial imbalances in the stratum corneum are known to cause skin allergies, psoriasis, acne and skin aging [32]. Keratinocytes play a key role in the responses of the body to the changing environment by controlling the production of hormones, neurotransmitters and cytokines. Changes in the skin microbiome due to infection by pathogenic or harmful bacteria or imbalance in the skin ecosystem can cause not only local skin problems but also other inflammatory diseases such as food allergies [33]. Bacterial skin infections are attributed to the invasion of bacteria through hair follicles or slightly damaged areas of the skin resulting from scratches, holes, surgery, burns, sunburns, animal or insect bites, wounds and conventional skin diseases. 

Since 2000, researchers have been focusing on the genetic factors causing AD and the relationship between the skin microbiome and AD. The skin microbiome seems to be one of the main causes of AD as an unbalanced skin ecosystem microbiome deteriorates the immune system as well as skin barriers. However, it is still unclear whether dysbiosis of the skin microbiome is the cause of the onset of AD or one of the symptoms of AD. Many other studies have shown low microbial diversity and abnormal microbial communities in skin tissues during inflammatory reactions in AD, regardless of age [6,34,35,36,37,38,39]. Chronic inflammation of the skin in AD patients led to reduced levels of *Cutibacterium*, *Streptococcus*, *Acinetobacter* and *Corynebacterium*, and subsequent increase in the strains of *Staphylococcus aureus*, which accounted for more than 90 % of the skin microbiome, of which more than 50–60 % were harmful [40,41,42]. In addition, the density of *S. aureus* is known to be closely related to the severity of AD regardless of the affected skin area [40,43]. The prevalence of AD among children is 20–30 %, whereas it is only 3 % in adults. This is because, in adults, the matured skin microbiome can potentially inhibit the growth of *S. aureus*, contributing to the age-related reduction in the incidence of AD. Interestingly, it was reported that certain strains of *S. aureus* isolated from the anterior nares of AD patients were able to form biofilm *in vitro*, which can be closely associated with the AD severity [43,44]. Chronic inflammation of the skin in AD patients led to reduced levels of *Corynebacterium* in the adult skin that theoretically contains genes involved in phosphoric metabolism that can reduce *S. aureus* infection by increasing competition among bacterial species [45]. Small molecules such as ribosomally synthesized and post-translationally modified peptides (RiPPs), glycolipids, terpenoids, non-ribosomal peptides (NRPs), polyketides, porphyrins and citrate amides are used by bacteria to interact with each other and their environment [46,47,48,49]. The genes required to produce these small molecules colocalize in biosynthetic gene clusters (BGCs). In the *in vitro* and mouse studies, adult skin microbes were found to secrete antibacterial metabolites such as small-molecule products of cutimycin from BGCs, which has been shown to inhibit the growth of *S. aureus* [50,51].

Until recently, most studies on skin microbial relevance to AD focused on *S. aureus* infection, with several studies showing that *Staphylococcus epidermidis*, a normal skin flora in humans, can inhibit the growth of *S. aureus* [40,52]. Skin colonization with *S. epidermidis* and *Staphylococcus hominis* reduced AD development in one-year-old children [26]. Some strains of *S. epidermidis* have also been found to improve innate immunity and activate IL-17-expressing CD8^+^ T cells to protect the skin from infection with pathogens [33]. *S. epidermidis* accounts for more than 90 % of the *Firmicutes* phylum in an anaerobic environment [50,53]. These bacteria have anti-inflammatory effects and help prevent the proliferation of pathogenic bacteria such as *S. aureus* strains, thereby acting as a skin barrier [4]. They also produce various bacteriocins—antibacterial peptides—to prevent the colonization and proliferation of pathogenic bacteria and the subsequent production of proinflammatory cytokines, and to strengthen the skin barrier to maintain skin homeostasis [54,55,56].

Skin barrier dysfunction caused by genetic mutations in the epidermal barrier-related genes such as filaggrin, and a family history of allergies are closely associated with the development of *S. aureus* colonization and AD [57]. A study on twins showed that a combination of genetic and environmental factors affects the skin microbiome and thus the function of the skin barrier [8,58]. Filaggrin is a filament-related protein that binds to keratin fibers in epidermal cells and is essential for the regulation of epidermal integrity. The skin of individual with a knockout mutation for the gene encoding filaggrin remains remarkably dry, making them vulnerable to skin conditions such as AD (eczema) and ichthyosis. The dysbiotic microbiome in the skin of AD patients promotes the generation of toxic factors that contribute to the severity of symptoms: the virulence of *S. aureus* includes α-toxin, which damages keratinocytes; δ-toxin, which stimulates mast cells; protein A, which triggers inflammatory responses from keratinocytes; superantigens, which trigger B cell expansion and cytokine release; and proinflammatory lipoproteins [59,60]. In addition, *S. aureus* expresses phenol-soluble modulins (PSMs), a family of peptides regulated by the accessory gene regulatory (agr) virulence system of *S. aureus* [61,62]. PSMs are encoded at three different locations in the genome and are tightly regulated by quorum sensing via the agr operon system. The PSMs produce two primary transcripts, i.e., RNAII and RNAIII. RNAII is generated by agrBDCA, an operon encoding factor necessary for the synthesis of autoinduction peptides (AIPs) and concentration-regulatory cascades. AIP, a peptide pheromone, is translated from AgrD, whereas AgrB transports the AIP into the extracellular space so that it can bind to the extracellular region of AgrC. AgrA and AgrC form a two-factor signal transduction system that regulates downstream signaling events, including RNAIII production that regulates viral factors, and includes an embedded hold gene sequence that generates δ-toxin [63]. Moreover, δ-toxin induces a decrease in mast cells of membrane-bound cytoplasmic granules containing histamine, IL-4 and IL-13, and induces the release of molecules important for activating immunoglobulin (Ig)E production in response to Th2-type skin inflammation [64,65]. These actions are accomplished by activating congenital immune receptors in immune cells, such as TLRs [24]. Thus, the development of the natural immune system of the skin can be attributed to the interaction of humans and microbiomes, which is an important factor for controlling the homeostasis of skin immunity [17].

The production of local immune substances influenced by microbes in the skin is not limited to the skin, but also affects the immunity of the entire body [66]. Exposure to rural environments with high levels of non-pathogenic bacteria, including *Acinetobacter*, is known to suppress allergic reactions in humans [67]. Exposure to germs from mothers during spontaneous delivery is also associated with a decrease in allergic reactions in children [33]. The administration of *Acinetobacter lwoffii*, isolated from rural areas in Germany, to pregnant animals through the nose could prevent an asthmatic phenotype, a type of allergic reaction, in offspring [68]. This effect is produced by type 1 T helper (Th1) cells in response to changes in the expression levels of IFN genes induced by exposure to bacterial species in the gut and brain. Injection of heat-treated *A. lwofii* into animal models was found to have a protective effect on allergic reaction and lung inflammation owing to Th1 cells and their anti-inflammatory effects [69]. These findings provide clear evidence that microbiota symbiosis in the skin can control the systemic immune response of the entire body.

## 3. The Role of the Gut Microbiome in Atopic Dermatitis

The gut microbiome is a microbial ecosystem in the intestine, and more than 90% of all microorganisms in humans are found in the large intestine [16,70]. The gut microbial community plays an essential role in the maturation of the human immune system, beginning at the first month of life and affecting individual immune responses. The acquisition of the gut microbiota begins at birth, after which it diversifies around six months after birth. During the third year of life, stability of the gut microbiota is achieved via development processes [17]. Metabolites and signal molecules such as RiPP, amino acid metabolites, acids (short-chain), oligosaccharides, glycolipids, and NRP produced by the gut microbiota can form a mucous layer in the gut and affect the systemic immune response of an individual [46,71]. Several related studies have shown that microorganisms in the intestine are related to chronic diseases ranging from gastrointestinal inflammatory and metabolic conditions to neurological, cardiovascular, and respiratory illnesses [72]. In particular, the brain–gut–microbiome axis theory suggests that the gut and brain are bidirectionally connected via the neuroendocrine system including hypothalamic-pituitary-adrenal axis and the vagus nerve, and closely interact with each other. A novel treatment strategy for dementia by using gut microbiome has been investigated: fecal microbiota transplantation (FMT) has been shown to have a positive effect on cognitive function in Alzheimer’s disease via alterations in the levels of circulating cytokines [73,74]. Gut microbiome functions as the first barrier to pathogenic microorganisms by adherence, producing substances that have antimicrobial effects, and stimulating immune responses in the host [75,76]. In addition, studies have been focusing on the gut microbiome as a future treatment therapy for AD, rheumatoid arthritis, diabetes and obesity [77].

The pathology of AD primarily includes immune anomalies and skin barrier defects. An imbalance in the gut microbiota is expected to play an important role in the etiological mechanism of AD. Several cohort studies have suggested that aberrant gut microbiota preceded the onset of AD, such as in infants with high fecal calprotectin levels (an antimicrobial protein used as a biomarker of intestinal inflammation) measured at 2 months of age who had an increased risk of AD and asthma by 6 years of age [7,73,78,79]. Some cohort studies have found that, in addition to small amounts of *Bifidobacterium* and *Bacteroides* and high levels of *Enterobacteriaceae*, infants with AD lack overall biological α-diversity [80,81].

In infants with AD, fewer species of the genus *Bacteroides* and many species of the genus *Firmicutes* were found. Certain species of *Bacteroides* genus, such as *Bacteroides thetaiotaomicron*, have an anti-inflammatory nature, which can be essential for alleviating allergic symptoms of AD as well as chronic inflammatory diseases including Crohn’s disease [82]. In addition, infants with AD, who as babies had skin tingling for 12 months, reported increased levels of *Clostridium* in the third week of life and decreased levels of *Bifidobacterium* in the bowel [80,81]. In other studies, infants with allergies were shown to carry a greater population of *Bifidobacterium adolescentis*, whereas, in healthy infants, *B. bifidum* was the dominant variant of the *Bifidobacterium* population [80]. In a preliminary large-scale cohort study, infants with AD were shown to have more colonization by *Clostridium difficile* and *Escherichia coli* than in infants without AD [83].

Conversely, studies have shown that infants with a high risk of allergic diseases having a diversity of gut microbiome have a low risk of AD [72,84,85]. The reduction of biodiversity in the gut microbiome and *Bacteroides* colonization in over one-month-old infants are related to AD [86]. Like in the skin, the diversity or differences among species in the diversity of the gut microbiome seem to be closely related to chronic skin diseases.

The immune system of the body and skin can be controlled via the gut microbiome as follows. The differentiation of naïve T cells to other types of Th cells such as Th1, Th2, Th17, or Foxp3^+^ Tregs depends remarkably on the condition of the gut microbiome. Tregs prevent dysfunctional T cells from differentiating into Th cells and inhibit the inflammatory activities of immune cells such as mast cells, eosinophils and basophils. Th cells also inhibit IgE production and induce IgG4 production [4,17]. *Bifidobacterium*, *Lactobacillus*, *Clostridium*, *Bacteroides, Streptococcus*, and their metabolites such as propionic and butyric acids are known for their ability to induce the polarization and expansion of Treg cells [87].

According to a recent study in Korea, significant dysbiosis (specifically an intraspecies compositional change) of a subspecies of *Faecalibacterium prausnitzii* was found in the fecal samples of patients with AD [9]. In addition, the simultaneous reduction of fecal SCFAs, such as butyrate and propionate, which have been known to exhibit anti-inflammatory effect and contribute to the maintenance of the integrity of epithelial barriers. Hence, a leaky gut in AD patients facilitates skin inflammation by allowing toxins, poorly digested foods, and gut microorganisms to penetrate the circulation of the body. When these reach the target tissue, including the skin, a strong Th2 reaction may be induced, causing further tissue damage [7,32,88]. SCFAs, such as acetate, propionate and butyrate, are the fermented products of dietary fibers in the gut, and are known to play an important role in determining the composition of the skin microbes, which are closely related to the skin’s immune defense mechanism [19,89]. For example, *Cutibacterium* produces acetate and propionate in the gut. Previous studies show that various skin conditions are known for their anti-inflammatory actions mediated by G-protein-coupled receptor 43 and for their contribution to epidermal barrier integrity. The anti-inflammatory activity is further mediated by Treg cells and induced by transforming growth factor (TGF)-β and/or IL-10. IL-10 exerts its inhibitory function by inducing TGF-β and other cytokines as well as suppressive signaling molecules, including cytotoxic T-lymphocyte antigen (CTLA)-4 and programmed death (PD)-1. In particular, propionic acid and its metabolized derivatives inhibit the growth of methicillin-resistant *S. aureus* USA300, a virulent strain that first emerged as community-associated methicillin-resistant *S. aureus* (MRSA) in the USA in the late 1990s [32]. Conversely, cutaneous bacteria such as *S. epidermidis* and *Cutibacterium acnes* (formerly *Propionibacterium acnes*) allow greater changes in SCFA production than others [90]. These findings suggest that an interaction occurs between the gut microbes and the skin.

## 4. Potential Pathways of the Gut–Skin Axis in Atopic Dermatitis

### 4.1. Mechanisms of How the Gut Microbiome Affects the Skin

Since dermatologists John H. Stokes and Donald M. Pillsbury first hypothesized that changes in the gut microbes could lead to skin inflammation causing conditions such as acne, several researches have supported the gut–brain–skin axis theory [1].

In patients with acne vulgaris, bacterial overgrowth in the small intestine was 10 times more common than in healthy controls [1]. Another study showed that patients with acne were more responsive to bacterial strains isolated from their stool [91]. Unlike control patients without skin disease, approximately 66 % of acne patients showed positive reactivity to stool-isolated coliform bacteria [1]. These results suggest that toxins from the gut microbes are a potential problem for patients with AD.

The gut–skin axis has been applied to not only simple skin diseases such as acne but also various chronic skin diseases with common symptoms, including AD (eczema). Thus, toxic substances from the gut microbiome affect and aggravate AD symptoms through particular pathways such as immunologic, metabolite and neuroendocrine pathway (Figure 1), and the underlying mechanisms need to be studied.

#### 4.1.1. Immunologic Pathway

In terms of immunity toward *S. aureus*, the most common bacterial strain affecting AD, a link was found between the gut and skin microbiotas. *S. aureus* is the most popular pathogen in the skin of AD patients. In contrast, according to a recent birth cohort study, colonization by the *S. aureus* strains played a role in the prevention of AD development in infancy, as early exposure to gut microorganisms is important at birth [8]. The exposure of the skin to *S. aureus*, similar to other cutaneous strains, facilitated the maturation of an infant’s immune system. *S. aureus* species in the skin may deteriorate the already established AD symptoms. However, before the occurrence of AD, the clustering of symbiotic *S. aureus* strains on the mucous membrane could exert protective effects through immunostimulation [8]. These studies support the possibility that the gut and skin are connected through the control of the immune environment via microbiome.

Specific gut microbes and their metabolites, such as retinoic acid and polysaccharide A from *Bacteroides fragilis*, *Faecalibacterium prausnitzii*, and bacterial species belonging to *Clostridium cluster IV* and *XI* promote the accumulation of Tregs and lymphocytes that stimulate anti-inflammatory reactions. Furthermore, some SCFAs, especially butyrate, regulate both the activation and apoptosis of immune cells [32].

An experiment performed using probiotics, including *Lactobacillus rhamnosus*, also verified the ameliorating effect of the gut microbiota on immunological skin irritation [32]. In that study, intestinal dysbiosis due to repeated water-avoiding stress was completely eliminated when the mice were treated with probiotics after exposure to stress [32].

#### 4.1.2. Metabolite Pathway

Metabolites, including SCFAs, produced by gut microbes or those further supplemented by oral administration also explain the association between the gut and skin via microbiome. SCFAs produced by gut microbes such as *Akkermansia muciniphila* play a key role in the etiology and pathology of AD, which can explain its association with the skin immune system. In mouse experiments, linoleic acid and 10-hydroxy-cis-12-octadecenoic acid mitigated AD symptoms and controlled the gut microbiome [92]. Furthermore, three different subgroups of neonatal gut microbiome (NGM1–3) and their metabolic products were shown to function in early allergic sensitization [93]. Of the three subgroups, NGM3 was related to multiple allergic sensitizations and was found to have a lower relative abundance of *Bifidobacterium*, *Akkermansia* and *Facalibacterium* [8]. For example, 12,13-dihydroxy-9Z-octadecenoic acid (12,13-DiHome), a metabolic product with an inflammatory effect on immune control *in vitro*, was rich in NGM3. Furthermore, 12,13-DiHome was increased in the protective layer of vernix caseosa, a white waxy coating found on newborn human skin. These findings may support the existence of a metabolite pathway in the gut– skin axis [8].

#### 4.1.3. Neuroendocrine Pathway

Like the skin, the lining of the gastrointestinal tract is also exposed to external environmental factors such as food and microorganisms. One of the main functions of the skin and gut is to inhibit the entry of any harmful pathogens, and the microbes on both organs can help eliminate these pathogens via immune function, and hence, establishing the stable microbiomes of both the organs and maintaining the proper balance is important for being healthy. Moreover, both microbiomes can affect each other through neuroendocrine signaling. This effect can occur via two routes: direct and indirect [8]. Tryptophan produced by intestinal microbes causing skin itching in AD patients is an example of direct signaling. In contrast, γ-aminobutyric acid produced by *Lactobacillus* and *Bifidobacterium* in the gut suppresses skin itching [8,94].

Through indirect channels, intestinal microbes regulate the concentration of cytokines such as IL-10 and IFN-γ in the bloodstream, which can lead to abnormal changes in brain function, resulting in anxiety and stress [8]. Cortisol, a representative stress hormone in humans, can alter the gut epithelium permeability and barrier function by changing the composition of the gut microbiota [95]. Cortisol can also change the levels of circulating neuroendocrine molecules such as tryptamine, trimethylamine and serotonin, thereby improving skin barrier and immune function [49].

In addition to these pathways, new evidence suggests that the gut microbiome and their metabolites affect skin microbes directly by being transferred to the skin. In the case of intestinal wall disorder, intestinal microbial metabolites access the bloodstream, accumulate in the skin, and can directly disrupt skin homeostasis [32].

### 4.2. Mechanisms of How the Skin Microbiome Affects the Gut

The extent to which skin microbes affect the immune system of the intestine has been investigated in the field of food allergy. Food allergy and the etiopathology of AD are closely related, in that atopic constitution is an important risk factor for food allergy outbreaks [96]. Thus, how the skin microbes affect the gut condition of AD patients can be determined. Exposure to food allergens through the skin barrier bypasses oral immunity. Thus, when the intestines are exposed to food allergens, prior experience of sensitization with the same allergens through the skin pathway leads to greater and effective food allergy-related skin symptoms such as itching. Epithelial sensitivity in damaged skin barriers is associated with the accumulation of thymic stromal lymphopoietin-induced basophils and dendritic cells that trigger antigen-induced food allergies. How skin exposure to microbes affects the immune sensitization and tolerance of the gut, similar to skin exposure to food allergens, is not yet known; further research is needed to determine whether this is related to the gut microbiome [32].

In the intestinal microbiota, bacteria prevail, and up to 100 species per individual are found; in contrast, skin microbes are dominated by fungi, viruses and rare bacteria, with about 40 species per individual [7]. Gut microbiomes are mostly acquired around six months after birth, and they undergo development until the age of two to six years. In contrast, skin microbes are acquired during infancy, but their colonization occurs once again during adolescence [7]. The manner in which the gut and skin microbiomes affect the development of AD is considerably similar. It involves induction of an immune response and inflammation and production of metabolites that are involved in damage to the skin barrier [7].

Although the manner in which the skin and gut microbiomes are involved in human immunity is similar, there are some differences. Intestinal microbes control the development of lymphatic structures related to the gut and play the most important role in activating congenital immunity. They can directly guide the gene regulatory network connectivity and control its function, stability and microbial-colonization resistance [97]. Skin microbes also control congenital immunity by generating antimicrobial peptides such as cathelicidin and β-defensin, by increasing the activity of the complement system, balancing the immune system between effective protection and damaging protection, and by controlling the level of IL-1 involved in the initiation and amplification of immune response. It also controls adaptive immunity by increasing IL-17A and IFN-fi from skin T cells and by controlling regulatory immunity, which eliminates exogenous pathogens. Colonization resistance of host microbiota occurs through bacteriocin, a serine protease; PSMs; SCFAs by *Cutibacterium* sp.; and porphyrin production by *S. epidermidis*. Thus, the collapse of the epidermal skin barrier by *S. aureus* colonization needs to be prevented in AD patients [7]. Although the exact mechanism by which this can be achieved is not yet known, injecting various skin-protective bacterial strains, including *S. epidermidis*, into AD patients might be an effective strategy.

A recent study showed that skin exposure to external elements can also affect the gut microbiome. Exposing the skin to narrow-band ultraviolet B (NB-UVB) increased serum vitamin D levels, leading to changes in the composition of the gut microbiome [98]. Although individuals taking vitamin D supplements did not show changes in gut bacterial composition, exposure to NB-UVB in participants without vitamin D supplementation increased the abundance of the species of the families *Lachnospiracheae, Rikenellaceae, Desulfobacteraceae, Clostridiales* vadinBB60 group, *Clostridia* Family XIII, *Coriobacteriaceae, Marinifilaceae*, and *Ruminococcaceae* [98]. This study showed that external elements could also affect the gut microbiome (Figure 1), and further studies are warranted to confirm this.

## 5. Perspective of Microbiome-Based Therapy for Atopic Dermatitis

### 5.1. Gut Microbiome-Targeted Therapies

#### 5.1.1. Probiotics

Most orally administered probiotics pass through the gastrointestinal tract that is generally hostile to survive, and is released after about a week while interacting with the gastrointestinal mucous membranes in which more than 70 % of immune cells are located [99]. Depending on the strains of probiotics, IL-12, IL-18 and tumor necrosis factor (TNF)-α can be generated to induce immune-stimulated signals or to stimulate the expression of anti-inflammatory cytokines such as IL-10 and TGF-β to generate immunity tolerance signals [100].

The combination of *L. rhamnosus* 19070-2 and *L. reuteri* DSM 122460 was used to manage AD, which had more pronounced effects in patients with positive skin terminal examination responses and increased IgE levels [101]. In a 12-week-long randomized, double-blind and placebo-controlled study performed in children aged one and twelve years, the use of *L. plantarum* CJLP133 strain reduced IFN-γ, eosinophil and IL-4 levels, and thus the AD scores such as SCORAD (SCORing Atopic Dermatitis) [102]. In another randomized, double-blind and placebo-controlled study, four months after the use of *L. paracasei*, *L. fermentum,* and their combination was discontinued in children, the SCORAD score of the group receiving probiotics was lower than that of the placebo group [103].

In addition, when rats were supplemented with *Lactobacillus brevis* SBC8803, the color of the cutaneous arterial sympathetic nerve became pale, and blood flow to the skin increased. This was probably because of the increased serotonin emission from intestinal cells and subsequent activation of parasympathetic neural pathways [32]. This effect was also reproduced in a human clinical study [32]. After *L. brevis* SBC8803 oral supplements were administered for 12 weeks to human subjects, the transepidermal water loss, an indicator of skin barrier function, decreased. In AD, skin barrier function is very important, and administration of oral probiotic supplements could help improve this function and thus improve AD symptoms [32].

Dysbiosis in the gut microbiome caused gluten sensitivity and low serum levels of vitamin D [104]. According to the skin–gut axis mentioned above, dysbiosis in the gut microbiome can cause AD; therefore, adjusting the blood levels of gluten or vitamin D can also be a treatment strategy for AD. Certain probiotics can hydrolyze gluten polypeptides, whereas others can increase the vitamin D level or further activate the expression of the vitamin D receptor [104,105].

The causes of AD and its resulting characteristics differ across individuals. Therefore, a uniform prescription of probiotics may have adverse effects or little efficacy. Thus, individual patients need to be analyzed to identify the cause and phenomenon of AD, and appropriate probiotics need to be prescribed (Figure 2).

#### 5.1.2. Pre- or Postbiotics

Recently, treatment of AD has involved not only probiotics but also postbiotics, the metabolites of probiotics, or prebiotics, the food of probiotics (Figure 2). The mechanism by which probiotics improve the symptoms of AD is by enhancing the production of SCFAs such as acetate, propionate and butyrate. SCFAs have anti-inflammatory effects, reduce the generation of toxic fermentation products, improve the Th1/Th2 ratio, increase the number of lymphocytes and/or leucocytes in the gut-associated lymphoid tissues, and increase intestinal IgA secretion [106]. A study found that prebiotic oligosaccharide mixtures added to infant diets before the first year of life reduced the average AD incidence by 44% [107].

Previous studies have shown that postbiotics suppress cell inflammation. Postbiotics are metabolites from intestinal bacteria; hence, they can function through the metabolite pathway from the gut to the skin, as mentioned previously [108,109]. LactoSporin^®^, an extracellular metabolite purified from *Bacillus coagulans* MTCC 5856 fermented broth (International Nomenclature Cosmetic Ingredient name, *Bacillus* ferment filtrate extract), was experimented as a postbiotic for the treatment of acne vulgaris. The potential mechanism of LactoSporin^®^ as an antimicrobial agent is through pH drop, microbial biofilm inhibition, and the draining of ions from the targeted cells [110]. As LactoSporin^®^ was used to treat acne vulgaris, it can also be used to treat AD. However, since the study was conducted in vitro, the exact mechanism underlying the effect of postbiotics and how they affect the systemic immune system of AD patients, especially that of the intestines, are not yet known. Postbiotic compounds of *Lactobacillus* were shown to have immune-modulating activities by increasing the number of Th1-related cytokines and decreasing Th2-related cytokines [111].

#### 5.1.3. Fecal Microbiota Transplantation

According to recent studies, Fecal Microbiota Transplantation (FMT) has become a popular means of intestinal microbiological control [112]. FMT refers to the transplantation of functional bacteria from the feces of healthy donors into the patient’s gastrointestinal tract to restore the balance of intestinal microorganisms and treat diseases related to gut microbial disorders. Previous studies show that various skin conditions appear to be potentially affected by the gut microbiome. In addition to hair loss and psoriasis, changes in acne and eczema in FMT recipients were observed [113,114,115,116,117]. There is also a recent study close to what we want to know, a cohort study involving a chart review of all patients who received FMT from January 2013 to December 2019 in a single academic medical center. However, it is not only an atopy-specific study but retrospectively limited to assessing whether the reported disease is affected or caused by the extended interval between FMT and dermatological visits. Therefore, further clinical trials and prospective studies are required to determine whether FMT is effective in AD patients. Of course, there is an ongoing clinical trial on the efficacy of FMT in adults with moderate-to-severe AD [118].

#### 5.1.4. Phage Therapy

Phage therapy refers to the treatment of certain bacterial infections by using bacteriophages that infect and kill the host bacteria. The advantage of using bacteriophages is that their host specificity; therefore, they do not have a significant impact on normal microbiomes of animals and plants. However, since phage therapy has not yet been clinically used for the treatment of AD although its safety is already fully guaranteed, and this review will only address this as a future possibility. As a member of the human gut microbiome, phages have recently been shown to be involved in human immune function through interactions with the intestinal microbiome [119].

Some studies have used phage therapy in a mice model of liver diseases because the internal microbiomes contain many *Enterococcus faecalis* that produce toxins (cytolysin) [120,121]. The findings of these studies might form the basis for using phage therapy to address the problems with the intestinal residence of harmful bacteria or the dysbiosis of the gut microbiome.

However, this therapy is associated with many uncertainties and limitations because whether the gut microbiome is the exact cause of AD is not yet known. All gut microbiome-based therapies have adverse effects and limitations. Overcoming these limitations in this field seems to be the greatest challenge in the future.

### 5.2. Skin Microbiome-Targeted Therapies

#### 5.2.1. Probiotics

The skin rash in AD patients is mainly due to the decrease in bacterial diversity because of the large volume of *S. aureus*; the standard treatments for AD usually improve skin symptoms by increasing bacterial diversity (Figure 2). Current treatments include local antibiotics (ointment), systemic antibiotics, and corticosteroids. The overuse of antibiotics can affect the natural skin microbiome as well as gut microbiome. If antibiotic overuse results in the generation and spread of drug-resistant bacteria, these bacteria can have a significant harmful impact on human health. This phenomenon can be especially fatal in subjects with an undeveloped immune system, such as newborns. Therefore, if antibiotics are needed for such vulnerable subjects, probiotics such as *Lactobacilli* along with prebiotics need to be administered together instead of the administration of antibiotics alone. In addition, bathing habits involving the excessive use of surfactants can deteriorate the balance of the skin microbiota.

Many researchers have attempted to identify the specific population of microbes that adversely affect AD and developed therapeutic methods to remove such populations. However, most of them suggested the use of probiotics to treat AD symptoms associated with gut microbiome dysbiosis [122,123]. For example, a probiotic strain, *Lactobacillus johnsonii*, showed significant improvement in skin symptoms of AD patients, which was attributed to the acidic metabolites of *Lactobacilli* that lower pH levels [122]. *Lactobacilli* are transmitted from the mother to the newborn; considering that these strains generally have a positive effect on the child’s immune system, their use for the treatment of AD is expected to have better outcomes [122].

Several studies have found significant improvement in specific skin conditions of subjects who take oral probiotics to change the gut microbiome rather than the skin microbiome. For example, *L. rhamnosus* GG reduces the sensitivity of AD patients with IgE-sensitive responses [43]. The growth of *S. aureus* can also be controlled by using *S. epidermis*, a commensal skin bacterium [124]. Several studies have shown that prebiotics such as glucomannan can prevent the proliferation of pathogenic bacteria and their spread in the skin, which can also help improve acne lesions and skin allergies. In addition, taking *Lactobacillus* alone can positively affect the skin (like skin microbes in the nasal barrier) [125,126].

Microorganisms from the feces can be extracted and transplanted into the skin. In animal models of human AD, microorganisms extracted from human feces were transplanted to strengthen the skin barrier and immune function [98]. *Vitreoscilla filiformis*, a non-pathogenic bacterium, was applied to skincare cosmetics to improve the skin microbial environment and function as well as to reduce the frequency of atopic eczema [39].

#### 5.2.2. Immunotherapy

Epicutaneous immunotherapy (EPIT) has been used not directly in AD but in the treatment of food allergies. EPIT is a new experimental method for transmitting low concentrations of food allergy-causing antigens through the healthy skin and inducing intrinsic resistance in patients with allergies [127]. In a mouse experiment, EPIT induced Tregs involved in the immune tolerance to milk allergy and promoted resistance to skin allergic reactions produced by peanuts and house dust mites [128]. Similarly, specific immunotherapy (SIT) was shown to have a significant positive effect on AD. SIT also showed significant efficacy in the long-term treatment of patients with severe AD [42]. As mentioned above, how relevant skin microbes are related to the etiopathology of AD as they are to food allergy is not yet known. Further studies are needed to determine whether the immune function of AD patients can be enhanced through stimulation by low concentrations of allergens.

## 6. Beyond the Bacterial Microbiome: Mycobiome and Virome in Atopic Dermatitis

The difference in skin fungal communities, “mycobiome” between patients with AD and healthy individuals was analyzed by using polymerase chain reaction (PCR). *Malassezia globosa* and *Malassezia restricta* were prevalent in all samples across both study groups, and some *Malassezia* species, including *Malassezia sloofiae* and *Malassezia dermatis*, had features of AD [129]. Additionally, there were significant differences in the virome composition of AD patients. While herpes simplex virus (HSV) showed similar relative abundance in both AD patients and healthy individuals, human papillomavirus (HPV) and bacteriophages were significantly increased in the skin of AD patients [130]. The reason why some mycobiome and virome are prevalent especially in the skin of AD is not known yet. However, it is sure that if the treatment of AD focusing on microbiota is not sufficient, we should consider mycobiome and virome as new supporting factors in the therapy.

## 7. Conclusions

Although interaction between the gut and skin microbiomes in AD have not been clearly demonstrated yet, several studies have implied that both microbiomes communicate directly or indirectly within the gut–skin axis, influencing the overall environment of the intestine and skin where they are located. The immunological, metabolic, and endocrine effects of the gut microbiome on the gut–skin axis have been revealed in detail. The oral administration of probiotics or manipulation of the microbiota itself can help in the treatment of AD. It is inferred that the skin microbiome can immunologically affect the gut microbiome through several studies, but still the influence of skin microbes on the gut microbiome and the underlying mechanism are not fully known yet. We could just identify the role of the skin microbes in food allergies related to AD pathology. This allowed us to assume that the skin microbiome plays a major role in the ‘gut–skin axis’ until now, despite the increasing importance of gut microbiome toward various organs. Furthermore, future studies need to focus on how intestinal and skin microbial management methods involving the use of probiotics, pre- or postbiotics, immunotherapy, phage therapy, and FMT work against AD compared to conventional treatments.

## Figures and Tables

**Figure 1 ijms-22-04228-f001:**
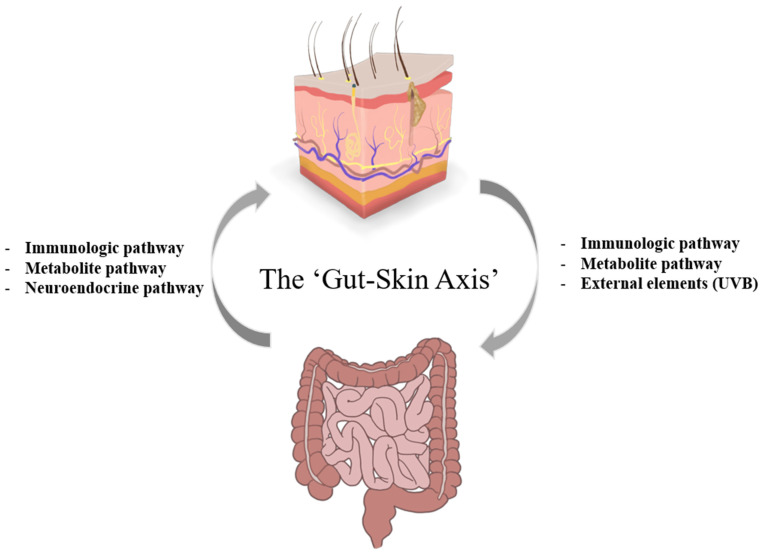
Several potential pathways of the gut–skin axis in atopic dermatitis (AD). In particular, numerous studies have shown that toxins in the gut microbiome are a potential problem for AD patients. Intestinal microbes are known to influence the skin through immunological, metabolic, and neuroendocrine pathways. The effect of the skin microbiome on the intestinal immune system has been mainly investigated in the field of food allergies, which shows how skin microorganisms affect the gut of AD patients. Recently, skin exposure to ultraviolet B (UVB) was shown to change the gut microbiome.

**Figure 2 ijms-22-04228-f002:**
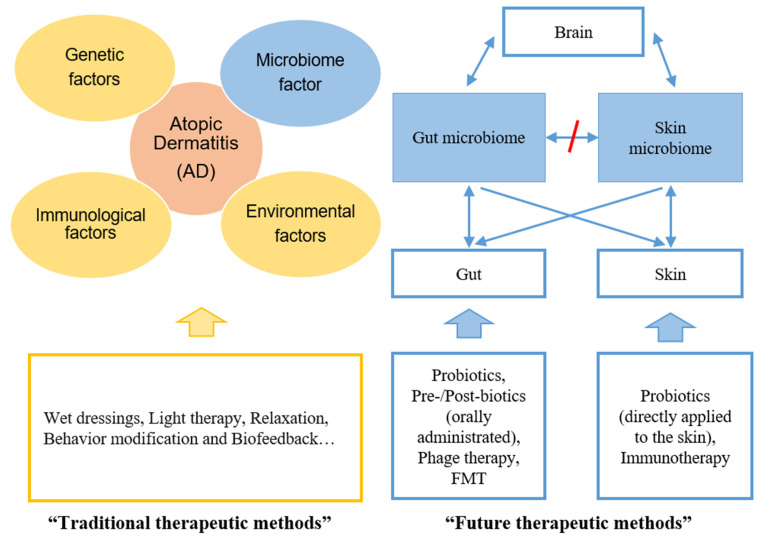
Comparisons between traditional (conventional) and future therapeutic methods for AD. Previous studies mainly focused on AD pathogenesis related to three factors: genetic, immunological, and environmental. Recently, a new treatment of AD targeting the microbiome has been proposed based on the gut–skin axis as the microbiome factor of the skin and gut. The gut microbiome-based treatments include oral administration of probiotics, prebiotics and postbiotics, and the possibility of using phage therapy. The skin microbiome-based treatments include direct application of probiotics to the skin and immunotherapy. Further studies are required to apply these microbiome-based methods as future treatments, instead of the existing treatments. FMT, fecal microbiota transplantation.

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
