# Peer review of "Comparative Analysis of the Microbiome across the Gut–Skin Axis in Atopic Dermatitis"

_ijms, 2021, doi:10.3390/ijms22084228_

Round 1

Reviewer 1 Report

The review summarizes the current knowledge regarding microbiome across the gut-skin axis and perspectives for future applications in clinical practice. I have the following minor suggestions:

  1. The introduction is too lengthy and should involve only brief information on atopic dermatitis and skin/gut microbiome (the title implies it is not the purpose of this review to discuss AD pathogenesis in detail). Please consider shortening the introduction.
  2. Please reference an article on the direct relationship between microbial dysbiosis of the gut and AD severity (lines 84-85); to my knowledge, such articles are lacking.
  3. Please reduce redundant sentences such as historical references to Stokes and Pillsbury's hypothesis (lines 93-94).
  4. Please consider using a more precise scientific language (e.g. lines 168-169 - what do you refer to as redness?, line 182 - destroys the immune system, 189 - toxic strains, i.e. toxin-producing? etc.).
  5. Why do you repeatedly differentiate between AD and eczema? Do you refer to contact dermatitis or some other entity?
  6. Alzheimer's disease and atopic dermatitis are abbreviated in the same way, i.e. AD - please change the abbreviation for the former.
  7. Please start listing the references in the text from 1, not from 2.
  8. Please remove the paragraph regarding the use of antibiotics in the treatment of microbial dysbiosis; based on current knowledge microbial dysbiosis without clinical signs of infection is not an indication for ATB therapy.
  9. Please consider adding more information about the microbial dysbiosis of the mucous membranes and the impact of bacterial biofilms in AD, e.g. based on the following references:

Nasal Colonization by Staphylococci and Severity of Atopic Dermatitis. Blicharz L, Usarek P, Młynarczyk G, Skowroński K, Rudnicka L, Samochocki Z. Dermatitis. 2020 May/Jun;31(3):215-222.

The Propensity to Form Biofilm in vitro by Staphylococcus aureus Strains Isolated from the Anterior Nares of Patients with Atopic Dermatitis: Clinical Associations. Blicharz L, Michalak M, Szymanek-Majchrzak K, Młynarczyk G, Skowroński K, Rudnicka L, Samochocki Z. Dermatology. 2020 Oct 28:1-7. doi: 10.1159/000511182.

10. Please correct some minor spelling mistakes (e.g. line 61 microbiol -->microbial).

Author Response

Reviewer 1

The review summarizes the current knowledge regarding microbiome across the gut-skin axis and perspectives for future applications in clinical practice. I have the following minor suggestions:

  1. The introduction is too lengthy and should involve only brief information on atopic dermatitis and skin/gut microbiome (the title implies it is not the purpose of this review to discuss AD pathogenesis in detail). Please consider shortening the introduction.

(Response) According to the reviewer’s comment, three paragraphs were deleted from the Introduction (lines 40-45, 68-75, 92-96, 101-117, highlighted in blue). 

  1. Please reference an article on the direct relationship between microbial dysbiosis of the gut and AD severity (lines 84-85); to my knowledge, such articles are lacking.

(Response) According to the reviewer’s comment, the expression of “It was well known that~” was changed to “It has been suggested that ~”. And several references such as 3, 6~9 were added.

  1. Please reduce redundant sentences such as historical references to Stokes and Pillsbury's hypothesis (lines 93-94).

(Response) According to the reviewer’s comment, that paragraph was deleted in the revised manuscript (line 92-96, highlighted in blue).

  1. Please consider using a more precise scientific language (e.g. lines 168-169 - what do you refer to as redness?, line 182 - destroys the immune system, 189 - toxic strains, i.e. toxin-producing? etc.).

(Response) According to the reviewer’s comment, “redness” was deleted from the line 178. And “destroys the immune system” was changed to “deteriorates the immune system” in the line 191, and “toxic” to “harmful” in the line 200.

  1. Why do you repeatedly differentiate between AD and eczema? Do you refer to contact dermatitis or some other entity?

(Response) According to the reviewer’s comment, “eczema” was deleted from the line 101, and changed to atopic dermatitis (AD) or put them in parenthesis as shown in the lines of 246, 333 and 412 in the revised manuscript.

  1. Alzheimer's disease and atopic dermatitis are abbreviated in the same way, i.e. AD - please change the abbreviation for the former.

(Response) According to the reviewer’s comment, “AD” meaning Alzheimer's disease in the line 315 was changed to “Alzheimer's disease” in the revised manuscript.

  1. Please start listing the references in the text from 1, not from 2.

(Response) According to the reviewer’s comment, reference in the text was started with “1” in the line 40 in the revised manuscript.

  1. Please remove the paragraph regarding the use of antibiotics in the treatment of microbial dysbiosis; based on current knowledge microbial dysbiosis without clinical signs of infection is not an indication for ATB therapy.

(Response) According to the reviewer’s comment, the paragraph regarding the antibiotics was deleted.

  1. Please consider adding more information about the microbial dysbiosis of the mucous membranes and the impact of bacterial biofilms in AD, e.g. based on the following references:

(Response) According to the reviewer’s comment, we added the paragraph describing the importance of biofilm-producing S. aureus on lesional skin (line 206-208), and the appropriate references (7, 43) in the revised manuscript. 

  1. Please correct some minor spelling mistakes (e.g. line 61 microbiol -->microbial).

(Response) According to the reviewer’s comment, “microbiol” was corrected to “microbial” (line 59).

Reviewer 2 Report

The review submitted by Dong Hoon Park and colleagues is generally well-written and understandable. The review explores the role of the skin and gut microbiota in atopic dermatitis, speculates on potential mechanisms, and discusses (microbiota-targeted) intervention studies. There are still quite a few concerns that I have, which are outlined below.

Major comments

  • The review links a lot of mechanisms together in a way that seems overly deterministic. For instance, AD is associated with reduced gut SCFA levels in one study. SCFAs are associated with reduced inflammatory markers in the circulation in the second study. AD is associated with inflammation in the third study. Hence, the reduction in colonic SCFAS causes inflammation in AD. However, to make such a “mechanistic” statement, one has to do mediator analysis within the same study, or a (preclinical) intervention study showing that SCFAs supplementation reduces inflammation and AD symptomatology in individuals with AD. Nonetheless, one can still speculate that such a mechanism for AD might be in place, but it needs to be framed in a more cautious way. The best example of this I could find is as follows - The role of the gut microbiome in AD: “propionic and butyric acids, are known for their ability to induce the polarization and expansion of Treg cells [95]. In other words, a leaky gut in AD patients facilitates skin inflammation by allowing toxins, poorly digested foods, and gut microorganisms to penetrate the circulation of the body.” I’m not sure how SCFA-induced changes in Treg cells equate intestinal permeability, subsequently causing skin inflammation by toxins entering the systemic inflammation.
  • After checking some of the references, they do not seem to support what is being said in the text, which is very worrying. For instance - 3.1.3. Fecal microbial transplantation: “Clinical trials have already been conducted on adult AD patients, and significant improvements in AD have been confirmed in the group undergoing FMT compared to the control group [118].” Reference 118 seems to be a link to a clinical study that hasn’t even been completed yet (https://clinicaltrials.gov/ct2/show/NCT04283968). The next sentence: “Numerous studies have suggested that FMT might play a very important role in treating AD by improving the intestinal microbial environment [119–121].” None of these studies even mention Atopic Dermatitis. These 2 sentences compromise 60% of the paragraph “3.1.3. Fecal microbial transplantation”
  • It is clear that there are still gaps in the literature that, when we uncover them, allow for more effective treatment methods for AD by targeting the microbiome. The conclusion talks about this to some extent, but it would be good to have a separate section on this really exploring what has been done and concrete next steps (i.e., we know some differences in the composition of the bacterial microbiome in AD, but we don’t know anything about phageome and mycobiome which seems like a next step to investigate; some probiotic studies seem promising for AD, but should/can these be improved? Is the treatment efficacy satisfactory? If not, how do we move forward etc.).
  • The review mentions the term “gut–brain–skin axis” a lot, but doesn’t actually explain too much about the concept. Please provide examples of how this axis (may) work using research articles.

Minor comments

  • The authors describe differences in the composition of the gut microbiota between conditions. Considering that both the gut and skin microbiota are discussed within the manuscript, please specify which type of microbiota whenever applicable.
  • Abstract: Incurable skin disease implies, to some extent, that it won’t be curable in the future. It might be more accurate to say that no known cure exists as of yet.
  • Abstract: Please use microbiota or microbiome instead of microflora. Microorganisms were initially placed in the Plant Kingdom, thus the term "Flora". Since microorganisms have been removed from the plant kingdom and are now in separate Kingdoms of their own, thus the tern "Flora" is incorrect.
  • Introduction: “In addition to genetic factors, environmental factors such as climate, air pollutants, diet, irritant exposure, and breastfeeding are responsible for AD outbreaks”. Responsible implies that all of these factors cause AD outbreaks, while some may be moderators or mediators. “are implicated in AD outbreaks” might be more accurate.
  • Introduction: “Hence, researchers have been attempting to identify the cause of AD in microbes in the skin and intestines”. It feels as if something is off in this sentence.
  • The role of skin microbiota in AD: “The skin microbiota consists of up to 107 microorganisms per square centimeter”. It seems like it should be 10^7, as opposed to 107?
  • The role of skin microbiota in AD: “The skin microbiome is the main cause of AD”, after which the following sentence describes associations between the skin microbiome and of AD. The authors should be more accurate when describing a causal link, versus findings that show an association. One of the following sentences shows something similar: “Chronic inflammation of the skin in AD patients led to reduced levels of Cutibacterium, Streptococcus, Acinetobacter etc”
  • The role of skin microbiota in AD: “in vitro” should be italicized.
  • The role of skin microbiota in AD: “Exposure to germs from mothers during pregnancy is also associated with a decrease in allergic reactions in children [46].” Does this imply that there is direct contact between the maternal microbiome and fetus during pregnancy in humans? If so, this seems quite controversial and might need some expansion. The concept of the placental microbiome is already somewhat controversial, so if this refers to the placental microbiome, then this needs to be acknowledged.
  • The role of the gut microbiome in AD: “the brain–gut–microbiome axis theory suggests that the gut and brain are connected via a microscopic boundary”. What is meant by microscopic boundary
  • The role of the gut microbiome in AD: “A novel treatment strategy for dementia by using gut microbiota has been investigated: fecal microbiota transplant (FMT) has been shown to have a positive effect on cognitive function in AD”. AD -> Alzheimer’s Disease is mentioned in this reference, while AD also refers to Atopic Dermatitis throughout the manuscript. If AD is used for both conditions, please change the abbreviation for one of them.
  • The role of the gut microbiome in AD: “such as in infants with high fecal calprotectin levels (an antimicrobial protein used as a biomarker of intestinal inflammation) measured at 2 months of age who had an increased risk of AD and asthma by 6 years of age [28,31,54,88]28,31,54,88.” The formatting for the references is off.
  • The role of the gut microbiome in AD: “infants with AD lack overall biological diversity [89,90].” Is alpha diversity meant? Beta diversity? Please specify.
  • The role of the gut microbiome in AD: what is meant by “The Bacteroides genus has an anti-inflammatory nature which is essential for alleviating AD symptoms.”? Bacteroides are one of the more prevalent genus of the microbiota and also contains pathogens.
  • The role of the gut microbiome in AD: “In addition, the simultaneous reduction of SCFAs contributed to the maintenance of the integrity of epithelial barriers and production of anti-inflammatory effects”. Please specify that this was part of the previous study [96]. Furthermore, this study did not measure intestinal permeability or inflammatory mediators, let alone perform a mediator analysis. So to say that “SCFAs contributed to the maintenance of the integrity of epithelial barriers and production of anti-inflammatory effects.” within this study seems like a stretch.
  • 1.3. Neuroendocrine pathway: “Tryptophan produced by intestinal microbes causing skin itching in AD patients is an example for direct signaling. In contrast, γ-aminobutyric acid produced by Lactobacillus and Bifidobacterium in the gut suppresses skin itching [70].” Please reference the original research article, as opposed to a review, whenever possible. Please also do so in the following sentence.
  • 1.1. Probiotics: The headings seem out of order.
  • 1.1. Probiotics: “Orally administered probiotics can adhere and reside in the gastrointestinal mucous membranes”. Many probiotics will just pass through the gastrointestinal tract without colonizing - https://www.sciencedirect.com/science/article/pii/S0966842X15000566
  • Conclusions: “No direct interactions have been shown to occur between the gut and skin microbiota in AD. A mechanistic study of the effects of gut and skin microbiota on AD pathology showed that each of them works independently.” Followed by “However, several studies have also shown that dermal microbes and intestinal microbiota interact indirectly within the gut–skin axis, influencing the overall environment of the intestine and skin where they are located.” Please add a reference of the original research articles showing this. In addition, this seems rather counter-intuitive. Because there is a special separation between the gut and the skin, it means by definition that any interaction between the 2 would be indirectly. As such, the first sentence implies that the skin and gut microbiota don’t interact with each other at all. However, such indirect interactions are depicted in figure 2. In addition, the following is said a few sentences later “the influence of skin microbes on the gut microbiome and the underlying mechanism are not yet known”, which is once again exactly the opposite to the first sentence of the conclusion.

Author Response

Reviewer II

The review submitted by Dong Hoon Park and colleagues is generally well-written and understandable. The review explores the role of the skin and gut microbiota in atopic dermatitis, speculates on potential mechanisms, and discusses (microbiota-targeted) intervention studies. There are still quite a few concerns that I have, which are outlined below.

<Major comments>

  1. The review links a lot of mechanisms together in a way that seems overly deterministic. For instance, AD is associated with reduced gut SCFA levels in one study. SCFAs are associated with reduced inflammatory markers in the circulation in the second study. AD is associated with inflammation in the third study. Hence, the reduction in colonic SCFAS causes inflammation in AD. However, to make such a “mechanistic” statement, one has to do mediator analysis within the same study, or a (preclinical) intervention study showing that SCFAs supplementation reduces inflammation and AD symptomatology in individuals with AD. Nonetheless, one can still speculate that such a mechanism for AD might be in place, but it needs to be framed in a more cautious way. The best example of this I could find is as follows - The role of the gut microbiome in AD: “propionic and butyric acids, are known for their ability to induce the polarization and expansion of Treg cells [95]. In other words, a leaky gut in AD patients facilitates skin inflammation by allowing toxins, poorly digested foods, and gut microorganisms to penetrate the circulation of the body.” I’m not sure how SCFA-induced changes in Treg cells equate intestinal permeability, subsequently causing skin inflammation by toxins entering the systemic inflammation.

(Response) According to the reviewer’s comment, a paragraph “In other words, a leaky gut in AD patients facilitates skin inflammation by allowing toxins, poorly digested foods, and gut microorganisms to penetrate the circulation of the body. When these reach the skin, a strong Th2 reaction may be induced, causing allergic reactions and subsequent tissue damage [16,54].” is deleted, and the paragraph of “Hence, a leaky gut in AD patients facilitates skin inflammation by allowing toxins, poorly digested foods, and gut microorganisms to penetrate the circulation of the body. When these reach target tissue, including the skin, a strong Th2 reaction may be induced, causing further tissue damage [7,32,87].” is added in line 371-375.

  1. After checking some of the references, they do not seem to support what is being said in the text, which is very worrying. For instance - 3.1.3. Fecal microbial transplantation: “Clinical trials have already been conducted on adult AD patients, and significant improvements in AD have been confirmed in the group undergoing FMT compared to the control group [118].” Reference 118 seems to be a link to a clinical study that hasn’t even been completed yet (https://clinicaltrials.gov/ct2/show/NCT04283968). The next sentence: “Numerous studies have suggested that FMT might play a very important role in treating AD by improving the intestinal microbial environment [119–121].” None of these studies even mention Atopic Dermatitis. These 2 sentences compromise 60% of the paragraph “3.1.3. Fecal microbial transplantation”

(Response) According to the reviewer’s comment, a paragraph “Clinical trials have already been conducted on adult AD patients, and significant improvements in AD have been confirmed in the group undergoing FMT compared to the control group [118]. Numerous studies have suggested that FMT might play a very important role in treating AD by improving the intestinal microbial environment [119–121]. However, most of the studies on FMT focus on improving the gut microbial environment; however, further clinical trials and studies are required to determine whether FMT is effective in AD patients.” is replaced with a paragraph “Previous studies show that there are various skin conditions that appear to be potentially affected by gut microbiome. In addition to hair loss and psoriasis, changes in acne and eczema in FMT recipients were observed [112-116]. There is also a recent study close to what we want to know, a cohort study involving chart review of all patients who received FMT from January 2013 to December 2019 in a single academic medical center. However, it is not only an atopy-specific study, but retrospectively limited to assessing whether the reported disease is affected or caused by the extended interval between FMT and dermatological visits. Therefore, further clinical trials and prospective studies are required to determine whether FMT is effective in AD patients. Of course, there is an ongoing clinical trial to efficacy of FMC in adults with moderate-to-severe AD [117].” As shown in the line 637-650.

  1. It is clear that there are still gaps in the literature that, when we uncover them, allow for more effective treatment methods for AD by targeting the microbiome. The conclusion talks about this to some extent, but it would be good to have a separate section on this really exploring what has been done and concrete next steps (i.e., we know some differences in the composition of the bacterial microbiome in AD, but we don’t know anything about phageome and mycobiome which seems like a next step to investigate; some probiotic studies seem promising for AD, but should/can these be improved? Is the treatment efficacy satisfactory? If not, how do we move forward etc.).

(Response) According to the reviewer’s comment, a paragraph about needs of further studies of mycobiome and human virome in the therapy of AD is added to the line 744-757 in the revised manuscript (highlighted in red).

  1. The review mentions the term “gut–brain–skin axis” a lot, but doesn’t actually explain too much about the concept. Please provide examples of how this axis (may) work using research articles.

(Response) According to the reviewer’s comment, a paragraph “The gut microbiome was found to act as a bridge between the immune system and the nervous system. In recent studies, in particular, this axis is used to describe the correlation between gut microbial communities, emotional states, systemic and skin inflammation, and may be associated with the mechanism between psoriasis and depression [14]. Using clinical cases of psoriasis and animal models, important communication pathways have been identified along the axis associated with the regulation of neurotransmitters in the microbiome [15]. It can be expected that a new strategy can be found to treat both psoriasis and depression based on the intestinal-brain-skin axis.” is added to line 117-127 in the revised manuscript (highlighted in red).

<Minor comments>

  • The authors describe differences in the composition of the gut microbiota between conditions. Considering that both the gut and skin microbiota are discussed within the manuscript, please specify which type of microbiota whenever applicable.

(Response) According to the reviewer’s comment, we additionally specify the types of microbiome, gut- or skin-, in the text where it is required (line 60, 91, 152 in the revised manuscript)

  • Abstract: Incurable skin disease implies, to some extent, that it won’t be curable in the future. It might be more accurate to say that no known cure exists as of yet.

(Response) According to the reviewer’s comment, “incurable” was changed to “refractory and relapsing” (line 17).

  • Abstract: Please use microbiota or microbiome instead of microflora. Microorganisms were initially placed in the Plant Kingdom, thus the term "Flora". Since microorganisms have been removed from the plant kingdom and are now in separate Kingdoms of their own, thus the tern "Flora" is incorrect.

(Response) According to the reviewer’s comment, “microflora” was changed to “microbiome” (line 21).

  • Introduction: “In addition to genetic factors, environmental factors such as climate, air pollutants, diet, irritant exposure, and breastfeeding are responsible for AD outbreaks”. Responsible implies that all of these factors cause AD outbreaks, while some may be moderators or mediators. “are implicated in AD outbreaks” might be more accurate.

(Response) According to the reviewer’s comment, “is responsible for” was changed to “is implicated in” (line 49)

  • Introduction: “Hence, researchers have been attempting to identify the cause of AD in microbes in the skin and intestines”. It feels as if something is off in this sentence.

(Response) According to the reviewer’s comment, that sentence was changed to “researchers have attempted to determine what microbes in the skin and intestines are related to AD and how they can be utilized to treat AD” (line 60-63).

  • The role of skin microbiota in AD: “The skin microbiota consists of up to 107 microorganisms per square centimeter”. It seems like it should be 10^7, as opposed to 107?

(Response) According to the reviewer’s comment, it was changed to “107” (line 160).

  • The role of skin microbiota in AD: “The skin microbiome is the main cause of AD”, after which the following sentence describes associations between the skin microbiome and of AD. The authors should be more accurate when describing a causal link, versus findings that show an association. One of the following sentences shows something similar: “Chronic inflammation of the skin in AD patients led to reduced levels of Cutibacterium, Streptococcus, Acinetobacter etc.”

(Response) According to the reviewer’s comment, “The skin microbiome is the main cause of AD” was changed to “The skin microbiome seems to be one of the main cause of AD” (line 190), and we also added the following sentence for clarifying the causal link the reviewer mentioned: “However, it is still unclear whether dysbiosis of skin microbiome is the cause of the onset of AD or one of the symptoms of AD” in the line 192-194.

  • The role of skin microbiota in AD: “in vitro” should be italicized.

(Response) “In vitro” was italicized in the line 207 and 218.

  • The role of skin microbiota in AD: “Exposure to germs from mothers during pregnancy is also associated with a decrease in allergic reactions in children [46].” Does this imply that there is direct contact between the maternal microbiome and fetus during pregnancy in humans? If so, this seems quite controversial and might need some expansion. The concept of the placental microbiome is already somewhat controversial, so if this refers to the placental microbiome, then this needs to be acknowledged.

(Response) According to the reviewer’s comment, “pregnancy” was changed to “spontaneous delivery” in the line 281.

  • The role of the gut microbiome in AD: “the brain–gut–microbiome axis theory suggests that the gut and brain are connected via a microscopic boundary”. What is meant by microscopic boundary

(Response) According to the reviewer’s comment, “a microscopic boundary” was changed to “the neuroendocrine systems including hypothalamic-pituitary-adrenal axis and the vagus nerve” in the line 310, 311.

  • The role of the gut microbiome in AD: “A novel treatment strategy for dementia by using gut microbiota has been investigated: fecal microbiota transplant (FMT) has been shown to have a positive effect on cognitive function in AD”. AD -> Alzheimer’s Disease is mentioned in this reference, while AD also refers to Atopic Dermatitis throughout the manuscript. If AD is used for both conditions, please change the abbreviation for one of them.

(Response) According to the reviewer’s comment, “AD” indicating Alzheimer’s disease was written in full name (line 315).

  • The role of the gut microbiome in AD: “such as in infants with high fecal calprotectin levels (an antimicrobial protein used as a biomarker of intestinal inflammation) measured at 2 months of age who had an increased risk of AD and asthma by 6 years of age [28,31,54,88]28,31,54,88.” The formatting for the references is off.

(Response) According to the reviewer’s comment, the superscript indication of the references “28,31,54,88. ” was deleted. (line 329)

  • The role of the gut microbiome in AD: “infants with AD lack overall biological diversity [89,90].” Is alpha diversity meant? Beta diversity? Please specify.

(Response) According to the reviewer’s comment, “overall biological diversity” was changed to “overall biological a-diversity” in the line 332.

  • The role of the gut microbiome in AD: what is meant by “The Bacteroides genus has an anti-inflammatory nature which is essential for alleviating AD symptoms.”? Bacteroides are one of the more prevalent genus of the microbiota and also contains pathogens.

(Response) According to the reviewer’s comment, that sentence was corrected to “Certain species of Bacteroides genus, such as Bacteroides thetaiotaomicron, has an anti-inflammatory nature, which can be essential for alleviating allergic symptoms of AD as well as chronic inflammatory diseases including Crohn’s disease”(line 334-337), and a related paper was added as reference (81. Delday, M.; Mulder, I.; Logan, E.T.; Grant, G. Bacteroides thetaiotaomicron Ameliorates Colon Inflammation in Preclinical Models of Crohn’s Disease. Inflamm. Bowel Dis. 2019, 25, 85-96). 

  • The role of the gut microbiome in AD: “In addition, the simultaneous reduction of SCFAs contributed to the maintenance of the integrity of epithelial barriers and production of anti-inflammatory effects”. Please specify that this was part of the previous study [96]. Furthermore, this study did not measure intestinal permeability or inflammatory mediators, let alone perform a mediator analysis. So to say that “SCFAs contributed to the maintenance of the integrity of epithelial barriers and production of anti-inflammatory effects.” within this study seems like a stretch.

(Response) According to the reviewer’s comment, that sentence was corrected to “the simultaneous reduction of fecal SCFAs, such as butyrate and propionate, which have been known to exhibit anti-inflammatory effect and contribute to the maintenance of the integrity of epithelial barriers.” (line 368-371)

  • 1.3. Neuroendocrine pathway: “Tryptophan produced by intestinal microbes causing skin itching in AD patients is an example for direct signaling. In contrast, γ-aminobutyric acid produced by Lactobacillus and Bifidobacterium in the gut suppresses skin itching [70].” Please reference the original research article, as opposed to a review, whenever possible. Please also do so in the following sentence.

(Response) According to the reviewer’s comment, the reference 70 was changed to the original article (8. Yokoyama, S.; Hiramoto, K.; Koyama, M.; Ooi, K. Impairment of skin barrier function via cholinergic signal transduction in a dextran sulphate sodium-induced colitis mouse model. Exp. Dermatol. 2015; 24: 779–784), and we also added another reference (93. Jin, U.H.; Lee, S.O.; Sridharan, G.; Lee, K.; Davidson, L.A.; Jayaraman, A.; Chapkin, R.S.; Alaniz, R.; Safe, S. Microbiome-derived tryptophan metabolites and their aryl hydrocarbon receptor-dependent agonist and antagonist activities. Mol. Pharmacol. 2014, 85, 777-788.)

  • 1.1. Probiotics: The headings seem out of order.

(Response) According to the reviewer’s comment, the heading numbers were all corrected.

  • 1.1. Probiotics: “Orally administered probiotics can adhere and reside in the gastrointestinal mucous membranes”. Many probiotics will just pass through the gastrointestinal tract without colonizing - https://www.sciencedirect.com/science/article/pii/S0966842X15000566

(Response) According to the reviewer’s comment, that sentence was changed to “Most of orally administered probiotics pass through the gastrointestinal tract that is generally hostile to survive and are released after about a week, while interacting with the gastrointestinal mucous membranes in which more than 70 % of immune cells are located.” (line 560-563) And the paper that the reviewer mentioned was added as reference.

  • Conclusions: “No direct interactions have been shown to occur between the gut and skin microbiota in AD. A mechanistic study of the effects of gut and skin microbiota on AD pathology showed that each of them works independently.” Followed by “However, several studies have also shown that dermal microbes and intestinal microbiota interact indirectly within the gut–skin axis, influencing the overall environment of the intestine and skin where they are located.” Please add a reference of the original research articles showing this. In addition, this seems rather counter-intuitive. Because there is a special separation between the gut and the skin, it means by definition that any interaction between the 2 would be indirectly. As such, the first sentence implies that the skin and gut microbiota don’t interact with each other at all. However, such indirect interactions are depicted in figure 2. In addition, the following is said a few sentences later “the influence of skin microbes on the gut microbiome and the underlying mechanism are not yet known”, which is once again exactly the opposite to the first sentence of the conclusion.

(Response) According to the reviewer’s comment, paragraph “No direct interactions have been shown to occur between the gut and skin microbiota in AD. A mechanistic study of the effects of gut and skin microbiota on AD pathology showed that each of them works independently. However, several studies have also shown that dermal microbes and intestinal microbiota interact indirectly within the gut–skin axis, influencing the overall environment of the intestine and skin where they are located.” is deleted (line 759-765, highlighted in blue) and a paragraph (line 769-774, highlighted in red) is revised in the revised manuscript.

Round 2

Reviewer 2 Report

The authors made substantial improvements to the manuscript. My opinion is that the manuscript is ready for publication.